# Provable Representation Learning for Imitation with Contrastive Fourier Features

**Ofir Nachum**
Google Brain
ofirnachum@google.com

**Mengjiao Yang**
Google Brain
sherryy@google.com

## Abstract

In imitation learning, it is common to learn a behavior policy to match an unknown *target policy* via max-likelihood training on a collected set of target demonstrations. In this work, we consider using offline experience datasets – potentially far from the target distribution – to learn low-dimensional state representations that provably accelerate the sample-efficiency of downstream imitation learning. A central challenge in this setting is that the unknown target policy itself may not exhibit low-dimensional behavior, and so there is a potential for the representation learning objective to alias states in which the target policy acts differently. Circumventing this challenge, we derive a representation learning objective that provides an upper bound on the performance difference between the target policy and a low-dimensional policy trained with max-likelihood, and this bound is tight *regardless* of whether the target policy itself exhibits low-dimensional structure. Moving to the practicality of our method, we show that our objective can be implemented as contrastive learning, in which the transition dynamics are approximated by either an implicit energy-based model or, in some special cases, an implicit *linear* model with representations given by random Fourier features. Experiments on both tabular environments and high-dimensional Atari games provide quantitative evidence for the practical benefits of our proposed objective.[1]

## 1 Introduction

In the field of sequential decision making one aims to learn a behavior policy to act in an environment to optimize some criteria. The well-known field of reinforcement learning (RL) corresponds to one aspect of sequential decision making, where the aim is to learn how to act in the environment to maximize cumulative returns via trial-and-error experience [38]. In this work, we focus on *imitation learning*, where the aim is to learn how to act in the environment to match the behavior of some unknown *target policy* [25]. This focus puts us closer to the supervised learning regime, and, indeed, a common approach to imitation learning – known as *behavioral cloning* (BC) – is to perform max-likelihood training on a collected a set of *target demonstrations* composed of state-action pairs sampled from the target policy [31, 34].

Since the learned behavior policy produces predictions (actions) conditioned on observations (states), the amount of demonstrations needed to accurately match the target policy typically scales with the state dimension, and this can limit the applicability of imitation learning to settings where collecting large amounts of demonstrations is expensive, , in health [18] and robotics [24] applications. The limited availability of target demonstrations stands in contrast to the recent proliferation of large *offline* datasets for sequential decision making [28, 19, 7, 21]. These datasets may exhibit behavior far from the target policy and so are not directly relevant to imitation learning via max likelihood

---

[1]Find experimental code at `https://github.com/google-research/google-research/tree/master/rl_repr`.

35th Conference on Neural Information Processing Systems (NeurIPS 2021).

training. Nevertheless, the offline datasets provide information about the unknown environment, presenting samples of environment reward and transition dynamics. It is therefore natural to wonder, is it possible to use such offline datasets to improve the sample efficiency of imitation learning?

Recent empirical work suggests that this *is* possible [40, 10], by using the offline datasets to learn a low-dimensional state representation via unsupervised training objectives. While these empirical successes are clear, the theoretical foundation for these results is less obvious. The main challenge in providing theoretical guarantees for such techniques is that of *aliasing*. Namely, even if environment rewards or dynamics exhibit a low-dimensional structure, the target policy and its demonstrations may not. If the target policy acts differently in states which the representation learning objective maps to the same low-dimensional representation, the downstream behavioral cloning objective may end up learning a policy which "averages" between these different states in unpredictable ways.

In this work, we aim to bridge the gap between practical objectives and theoretical understanding. We derive an offline objective that learns low-dimensional representations of the environment dynamics and, if available, rewards. We show that minimizing this objective in conjunction with a downstream behavioral cloning objective corresponds to minimizing an upper bound on the performance difference between the learned low-dimensional BC policy and the unknown and possibly high-dimensional target policy. The form of our bound immediately makes clear that, as long as the learned policy is sufficiently expressive on top of the low-dimensional representations, the implicit "averaging" occurring in the BC objective due to any aliasing is irrelevant, and a learned policy can match the target regardless of whether the target policy itself is low-dimensional.

Extending our results to policies with limited expressivity, we consider the commonly used parameterization of setting the learned policy to be log-linear with respect to the representations (, a softmax of a linear transformation). In this setting, we show that it is enough to use the same offline representation learning objective, but with linearly parameterized dynamics and rewards, and this again leads to an upper bound showing that the downstream BC policy can match the target policy *regardless* of whether the target is low-dimensional or log-linear itself. We compare the form of our representation learning objective to "latent space model" approaches based on bisimulation principles, popular in the RL literature [20, 41, 22, 13], and show that these objectives are, in contrast, very liable to aliasing issues even in simple scenarios, explaining their poor performance in recent empirical studies [40].

We continue to the practicality of our own objective, and show that it can be implemented as a contrastive learning objective that implicitly learns an energy based model, which, in many common cases, corresponds to a *linear* model with respect to representations given by random Fourier features [33]. We evaluate our objective in both tabular synthetic domains and high-dimensional Atari game environments [11]. We find that our representation learning objective effectively leverages offline datasets to dramatically improve performance of behavioral cloning.

## 2   Related Work

Representation learning in sequential decision making has traditionally focused on learning representations for improved RL rather than imitation. While some works have proposed learning *action* representations [8, 30], our work focuses on *state* representation learning, which is more common in the literature, and whose aim is generally to distill aspects of the observation relevant to control from those relevant only to measurement [3]; see [27] for a review. Of these approaches, bisimulation is the most theoretically mature [17, 13], and several recent works apply bisimulation principles to derive practical representation learning objectives [20, 41, 6]. However, the existing theoretical results for bisimulation fall short of the guarantees we provide. For one, many of the bisimulation results rely on defining a representation error which holds *globally* on all states and actions [1]. On the other hand, theoretical bisimulation results that define a representation error in terms of an *expectation* are inapplicable to imitation learning, as they only provide guarantees bounding the performance difference between policies that are "close" (, Lipschitz) in the representation space and say nothing regarding whether an arbitrary target policy in the true MDP can be represented in the latent MDP [20]. In Section 4.4, we will show that these shortcomings of bisimulation fundamentally limit its applicability to an imitation learning setting.

In contrast to RL, there are comparatively fewer theoretical works on representation learning for imitation learning. One previous line of research in this vein is given by [9], which considers learning a state representation using a dataset of multiple demonstrations from multiple target policies.

Accordingly, this approach requires that each target policy admits a low-dimensional representation. In contrast, our own work makes no assumption on the form of the target policy, and, in fact, this is one of the central challenges of representation learning in this setting.

As imitation learning is close to supervised learning, it is an interesting avenue for future work to extend our results to more common supervised learning domains. We emphasize that our own contrastive objectives are distinct from typical approaches in image domains [15] and popular in image-based RL [26, 37, 36], which use prior knowledge to generate pairs of similar images (, via random cropping). We avoid any such prior knowledge of the task, and our losses are closer to temporal contrastive learning, more common in NLP [29].

## 3 Background

We begin by introducing the notation and concepts we will build upon in the later sections.

**MDP Notation** We consider the standard MDP framework [32], in which the environment is given by a tuple $\mathcal{M} := \langle S, A, \mathcal{R}, \mathcal{P}, \mu, \gamma \rangle$, where $S$ is the state space, $A$ is the action space, $\mathcal{R} : S \times A \to [-R_{\max}, R_{\max}]$ is the reward function, $\mathcal{P} : S \times A \to \Delta(S)$ is the transition function,[2] $\mu \in \Delta(S)$ is the initial state distribution, and $\gamma \in [0, 1)$ is the discount factor. In this work, we restrict our attention to finite action spaces, $|A| \in \mathbb{N}$. A stationary policy in this MDP is a function $\pi : S \to \Delta(A)$. A policy acts in the environment by starting at an initial state $s_0 \sim \mu$ and then at time $t \geq 0$ sampling an action $a_t \sim \pi(s_t)$. The environment then produces a reward $r_t = \mathcal{R}(s_t, a_t)$ and stochastically transitions to a state $s_{t+1} \sim \mathcal{P}(s_t, a_t)$. While we consider deterministic rewards, all of our results readily generalize to stochastic rewards, in which case the same bounds hold for $\mathcal{R}(s, a)$ denoting the expected value of the reward at $(s, a)$.

The *performance* associated with a policy $\pi$ is its expected future discounted reward when acting in the manner described above:

$$J_{\mathrm{Perf}}(\pi) := \mathbb{E}\left[ \sum_{t=0}^{\infty} \gamma^t \mathcal{R}(s_t, a_t) \ \middle| \ \pi, \mathcal{M} \right]. \tag{1}$$

The *visitation distribution* of $\pi$ is the state distribution induced by the sequential process:

$$d^{\pi}(s) := (1 - \gamma) \sum_{t=0}^{\infty} \gamma^t \cdot \mathrm{Pr}\left[ s_t = s | \pi, \mathcal{M} \right]. \tag{2}$$

**Behavioral Cloning (BC)** In imitation learning, one wishes to recover an unknown *target* policy $\pi_*$ with access to demonstrations of $\pi_*$ acting in the environment. More formally, the demonstrations are given by a dataset $\mathcal{D}_N^{\pi_*} = \{(s_i, a_i)\}_{i=1}^N$ where $s_i \sim d^{\pi_*}, a_i \sim \pi_*(s_i)$. A popular approach to imitation learning is *behavioral cloning* (BC), which suggests to learn a policy $\pi$ to approximate $\pi_*$ via max-likelihood optimization. That is, one wishes to use the $N$ samples to approximately minimize the objective

$$J_{\mathrm{BC}}(\pi) := \mathbb{E}_{(s,a) \sim (d^{\pi_*}, \pi_*)}[-\log \pi(a|s)]. \tag{3}$$

In this work, we consider using a state representation function to simplify this objective. Namely, we consider a function $\phi : S \to Z$. Given this representation, one no longer learns a policy $\pi : S \to \Delta(A)$, but rather a policy $\pi_Z : Z \to \Delta(A)$. The BC loss with representation $\phi$ becomes

$$J_{\mathrm{BC},\phi}(\pi_Z) := \mathbb{E}_{(s,a) \sim (d^{\pi_*}, \pi_*)}[-\log \pi_Z(a|\phi(s))]. \tag{4}$$

A smaller representation space $Z$ can help reduce the hypothesis space for $\pi_Z$ compared to $\pi$, and this in turn reduces the number of demonstrations $N$ needed to achieve small error in $J_{\mathrm{BC},\phi}$. However, whether a small error in $J_{\mathrm{BC},\phi}$ translates to a small error in $J_{\mathrm{BC}}$ depends on the nature of $\phi$, and thus how to determine a good $\phi$ is a central challenge.

**Offline Data** In this work, we consider learning $\phi$ via offline objectives. We assume access to a dataset of transition tuples $\mathcal{D}_M^{\mathrm{off}} = \{(s_i, a_i, r_i, s_i')\}_{i=1}^M$ sampled independently according to

$$s_i \sim d^{\mathrm{off}}, a_i \sim \mathrm{Unif}_A, r_i = \mathcal{R}(s_i, a_i), s_i' \sim \mathcal{P}(s_i, a_i), \tag{5}$$

---

[2]We use $\Delta(\mathcal{X})$ to denote the simplex over a set $\mathcal{X}$.

where $d^{\text{off}}$ is some unknown offline state distribution. We assume that the support of $d^{\text{off}}$ includes the support of $d^{\pi_*}$; , $d^{\pi_*}(s) > 0 \Rightarrow d^{\text{off}}(s) > 0$. The uniform sampling of actions in $\mathcal{D}^{\text{off}}$ follows similar settings in related work [5] and in principle can be replaced with any distribution uniformly bounded from below by $\eta > 0$ and scaling our derived bounds by $\frac{1}{|A|\eta}$. At times we will abuse notation and write samples of these sub-tuples as $(s, a) \sim d^{\text{off}}$ or $(s, a, r, s') \sim d^{\text{off}}$.

**Learning Goal** Similar to related work [9], we will measure the discrepancy between a candidate $\pi$ and the target $\pi_*$ via the *performance difference*:

$$\text{PerfDiff}(\pi, \pi_*) := |J_{\text{Perf}}(\pi) - J_{\text{Perf}}(\pi_*)|. \tag{6}$$

At times we will also use the notation $\text{PerfDiff}(\pi_Z, \pi_*)$, and this is understood to mean $\text{PerfDiff}(\pi_Z \circ \phi, \pi_*)$. While we focus on the performance difference, all of our results may be easily modified to alternative evaluation metrics based on distributional divergences, , $D_{\text{TV}}(d^{\pi_*} \| d^\pi)$ or $D_{\text{KL}}(d^{\pi_*} \| d^\pi)$, where $D_{\text{TV}}$ is the total variation (TV) divergence and $D_{\text{KL}}$ is the Kullback Leibler (KL) divergence. Also note that we make no assumption that $\pi_*$ is an optimal or near-optimal policy in $\mathcal{M}$.

In the case of vanilla behavioral cloning, we have the following relationship between $J_{\text{BC}}$ and the performance difference, which is a variant of Theorem 2.1 in [34].

**Lemma 1.** *For any $\pi, \pi_*$, the performance difference may be bounded as*

$$\text{PerfDiff}(\pi, \pi_*) \leq \frac{R_{\max}}{(1 - \gamma)^2} \sqrt{2 \mathbb{E}_{d^{\pi_*}} [D_{\text{KL}}(\pi_*(s) \| \pi(s))]} = \frac{R_{\max}}{(1 - \gamma)^2} \sqrt{\text{const}(\pi_*) + 2 J_{\text{BC}}(\pi)}. \tag{7}$$

*See Appendices B and C for all proofs.*

**Remark** (Quadratic dependence on horizon)**.** *Notice that the guarantee above for vanilla BC includes a quadratic dependence on horizon in the form of $(1 - \gamma)^{-2}$, and this quadratic dependence is maintained in all our subsequent bounds. While there exists a number of imitation learning works that aim to reduce this dependence, the specific problem our paper focuses on – aliasing in the context of learning state representations – is an orthogonal problem to quadratic dependence on horizon. Indeed, if some representation maps two very different raw observations to the same latent state, no downstream imitation learning algorithm (regardless of sample complexity) will be able to learn a good policy. Still, extending our representation learning bounds to more sophisticated algorithms with potentially smaller dependence on horizon, like DAgger [35], is a promising direction for future work.*

## 4 Representation Learning with Performance Bounds

We now continue to our contributions, beginning by presenting performance difference bounds analogous to Lemma 1 but with respect to a specific representation $\phi$. The bounds will necessarily depend on quantities which correspond to how "good" the representation is, and these quantities then form the representation learning objective for learning $\phi$; ideally these quantities are independent of $\pi_*$, which is unknown.

Intuitively, we need $\phi$ to encapsulate the important aspects of the environment. To this end, we consider representation-based models $\mathcal{P}_Z : Z \times A \to \Delta(S)$ and $\mathcal{R}_Z : Z \times A \to [-R_{\max}, R_{\max}]$ of the environment transitions and rewards. We define the error incurred by these models on the offline distribution as,

$$J_{\text{R}}(\mathcal{R}_Z, \phi)^2 := \mathbb{E}_{(s,a) \sim d^{\text{off}}} \left[ (\mathcal{R}(s, a) - \mathcal{R}_Z(\phi(s), a))^2 \right], \tag{8}$$

$$J_{\text{T}}(\mathcal{P}_Z, \phi)^2 := \frac{1}{2} \mathbb{E}_{(s,a) \sim d^{\text{off}}} \left[ D_{\text{KL}}(\mathcal{P}(s, a) \| \mathcal{P}_Z(\phi(s), a)) \right]. \tag{9}$$

We will elaborate on how these errors are translated to representation learning objectives in practice in Section 5, but for now we note that the expectation over $d^{\text{off}}$ already leads to a connection between theory and practice much closer than exists in other works, which often resort to supremums over state-action errors [30, 41, 1] or zero errors globally [17].

### 4.1 General Policies

We now relate the representation errors above to the performance difference, analogous to Lemma 1 but with $J_{\mathrm{BC},\phi}$. We emphasize that the performance difference below measures the difference in returns in the *true* MDP $\mathcal{M}$, rather than, as is commonly seen in model-based RL [23], the difference in the latent MDP defined by $\mathcal{R}_Z, \mathcal{P}_Z$.

**Theorem 2.** *Consider a representation function $\phi : S \to Z$ and models $\mathcal{R}_Z, \mathcal{P}_Z$ as defined above. Denote the representation error as*

$$\epsilon_{\mathrm{R,T}} := \frac{|A|}{1-\gamma} J_{\mathrm{R}}(\mathcal{R}_Z, \phi) + \frac{2\gamma|A|R_{\max}}{(1-\gamma)^2} J_{\mathrm{T}}(\mathcal{P}_Z, \phi). \tag{10}$$

*Then the performance difference in $\mathcal{M}$ between $\pi_*$ and a latent policy $\pi_Z : Z :\to \Delta(A)$ may be bounded as,*

$$\mathrm{PerfDiff}(\pi_Z, \pi_*) \leq \underbrace{(1 + D_{\chi^2}(d^{\pi_*}\|d^{\mathrm{off}})^{\frac{1}{2}}) \cdot \epsilon_{\mathrm{R,T}}}_{\textit{offline pretraining}} + C\sqrt{\underbrace{\frac{1}{2}\mathbb{E}_{z\sim d_Z^{\pi_*}}[D_{\mathrm{KL}}(\pi_{*,Z}(z)\|\pi_Z(z))]}_{= \underbrace{\mathrm{const}(\pi_*,\phi) + J_{\mathrm{BC},\phi}(\pi_Z)}_{\textit{downstream behavioral cloning}}}}, \tag{11}$$

*where $C = \frac{2R_{\max}}{(1-\gamma)^2}$ and $d_Z^{\pi_*}, \pi_{*,Z}$ are the marginalization of $d^{\pi_*}, \pi_*$ onto $Z$ according to $\phi$:*

$$d_Z^{\pi_*}(z) := \Pr[z = \phi(s) \mid s \sim d^{\pi_*}] ; \quad \pi_{*,Z}(z) := \mathbb{E}[\pi_*(s) \mid s \sim d^{\pi_*}, z = \phi(s)]. \tag{12}$$

*Proof in Appendix B.2.*

We thus have a guarantee showing that $\pi_Z$ can match the performance of $\pi_*$ *regardless* of the form of $\pi_*$ (, whether $\pi_*$ itself possesses a low-dimensional parameterization). Indeed, as long as $\phi$ is is learned well enough ($\epsilon_{\mathrm{R,T}} \to 0$) and the space of candidate $\pi_Z$ is expressive enough, optimizing $J_{\mathrm{BC},\phi}$ can achieve near-zero performance difference, since the latent policy optimizing $J_{\mathrm{BC},\phi}$ is $\pi_Z = \pi_{*,Z}$, and this setting of $\pi_Z$ zeros out the second term of the bound in (11). Note that, in general $\pi_* \neq \pi_{*,Z} \circ \phi$, yet the performance difference of these two distinct policies is nevertheless zero when $\epsilon_{\mathrm{R,T}} = 0$. The bound in Theorem 2 also clearly exhibits the trade-off due to the offline distribution, encapsulated by the Pearson $\chi^2$ divergence $D_{\chi^2}(d^{\pi_*}\|d^{\mathrm{off}})$.

**Remark** (Representations agnostic to environment rewards)**.** *In some imitation learning settings, rewards are unobserved and so optimizing $J_{\mathrm{R}}$ is infeasible. In these settings, one can consider setting $\mathcal{R}_Z$ to an (unobserved) constant function $\mathbb{E}_{d^{\mathrm{off}}}[\mathcal{R}(s,a)]$, ensuring $J_{\mathrm{R}}(\mathcal{R}_Z, \phi) \leq R_{\max}$. Furthermore, in environments where the reward does not depend on the action, , $\mathcal{R}(s, a_1) = \mathcal{R}(s, a_2) \forall a_1, a_2 \in A$, the bound in Theorem 2 can be modified to remove $J_{\mathrm{R}}$ altogether (see Appendix B for details).*

### 4.2 Log-linear Policies

Theorem 2 establishes a connection between the performance difference and behavioral cloning over representations given by $\phi$. The optimal latent policy for BC is $\pi_{*,Z}$, and this is the same policy which achieves minimal performance difference. Whether we can find $\pi_Z \approx \pi_{*,Z}$ depends on how we parameterize our latent policy. If $\pi_Z$ is tabular or if $\pi_Z$ is represented as a sufficiently expressive neural network, then the approximation error is effectively zero. But what about in other cases?

In this subsection, we consider $Z \subset \mathbb{R}^d$ and log-linear policies of the form

$$\pi_\theta(z) = \mathrm{softmax}(\theta^\top z) := \left(\frac{\exp\{\theta_a^\top z\}}{\sum_{\tilde{a}} \exp\{\theta_{\tilde{a}}^\top z\}}\right)_{a \in A}, \tag{13}$$

where $\theta \in \mathbb{R}^{d \times |A|}$. In general, $\pi_{*,Z}$ cannot be expressed as $\pi_\theta$ for some $\theta$. Nevertheless, we can still derive strong bounds for this scenario, by considering factored *linear* [4] models $\mathcal{R}_Z, \mathcal{P}_Z$:

**Theorem 3.** *Consider $Z \subset \mathbb{R}^d$, a representation function $\phi : S \to Z$, and linear models $\mathcal{R}_Z, \mathcal{P}_Z$:*

$$\mathcal{R}_Z(z,a) := r(a)^\top z ; \quad \mathcal{P}_Z(s'|z,a) := \psi(s',a)^\top z \text{ for some } r : A \to \mathbb{R}^d ; \psi : S \times A \to \mathbb{R}^d.$$

*Denote the representation error $\epsilon_{\mathrm{R,T}}$ as in Theorem 2. Then the performance difference between $\pi_*$ and a latent policy $\pi_\theta(z) := \mathrm{softmax}(\theta^\top z)$ may be bounded as,*

$$\mathrm{PerfDiff}(\pi_\theta, \pi_*) \leq (1 + D_{\chi^2}(d^{\pi_*}\|d^{\mathrm{off}})^{\frac{1}{2}}) \cdot \epsilon_{\mathrm{R,T}} + C \cdot \left\|\frac{\partial}{\partial\theta} J_{\mathrm{BC},\phi}(\pi_\theta)\right\|_1, \tag{14}$$

where $C = \frac{1}{1-\gamma}\|r\|_\infty + \frac{\gamma R_{\max}}{(1-\gamma)^2}\|\psi\|_\infty$.

*Proof in Appendix B.3.*

The statement of Theorem 3 makes it clear that realizability of $\pi_{*,Z}$ is irrelevant for log-linear policies. It is enough to only have the gradient with respect to learned $\theta$ be close to zero, which is a guarantee of virtually all gradient-based algorithms. Thus, in these settings performing BC on top of learned representations is provably optimal *regardless* of both the form of $\pi_*$ and the form of $\pi_{*,Z}$.

**Remark** (Kernel-based models). *It is possible to extend the statement of Theorem 3 to generalized linear dynamics and reward models based on kernels by replacing the gradient in the bound with the* functional gradient *with respect to the kernel [16].*

### 4.3 Sample Efficiency

The previous theorems show that we can reduce imitation learning to (1) representation learning on an offline dataset, and (2) behavioral cloning on target demonstrations with the learned representation. How does this compare to performing BC on the target demonstrations directly? Intuitively, representation learning should help when the learned representation is "good" (, $\epsilon_{R,T}$ is small) and the complexity of the representation space $Z$ is low (, $|Z|$ is small or $Z \subset \mathbb{R}^d$ for small $d$). In this subsection, we formalize this intuition for a simple setting. We consider finite $S, Z$. For the representation $\phi$, we assume access to an oracle $\phi_M := \mathcal{OPT}_\phi(\mathcal{D}_M^{\mathrm{off}})$ which yields an error $\epsilon_{R,T}(\phi_M)$. For BC we consider learning a tabular $\pi_Z$ on the finite demonstration set $\mathcal{D}_N^{\pi_*}$. We have the following theorem, which characterizes the expected performance difference when using representation learning.

**Theorem 4.** *Consider the setting described above. Let $\phi_M := \mathcal{OPT}_\phi(\mathcal{D}_M^{\mathrm{off}})$ and $\pi_{N,Z}$ be the policy resulting from BC with respect to $\phi_M$. Then we have,*

$$\mathbb{E}_{\mathcal{D}_N^{\pi_*}}[\mathrm{PerfDiff}(\pi_{N,Z}, \pi_*)] \leq (1 + D_{\chi^2}(d^{\mathrm{off}}\|d^{\pi_*})^{\frac{1}{2}}) \cdot \epsilon_{R,T}(\phi_M) + C \cdot \sqrt{\frac{|Z||A|}{N}}, \quad (15)$$

*where $C$ is as in Theorem 2.*

*Proof in Appendix C.1.*

Note that application of vanilla BC to this setting would achieve a similar bound but with $\epsilon_{R,T} = 0$ and $|Z| = |S|$. Thus, an improvement from representation learning is expected when $\epsilon_{R,T}(\phi_M)$ and $|Z|$ are small.

### 4.4 Comparison to Bisimulation

The form of our representation learning objectives – learning $\phi$ to be predictive of rewards and next state dynamics – recalls similar ideas in the bisimulation literature [17, 20, 41, 13]. However, a key difference is that in bisimulation the divergence over next state dynamics is measured in the latent representation space; , a divergence between $\phi \circ \mathcal{P}(s, a)$ and $f(\phi(s), a)$ for some "latent space model" $f$, whereas our proposed representation error is between $\mathcal{P}(s, a)$ and $\mathcal{P}_Z(\phi(s), a)$. We find that this difference is crucial, and in fact there exist no theoretical guarantees for bisimulation similar to those in Theorems 2 and 3. Indeed, one can construct a simple example where the use of latent space models leads to a complete failure, see Figure 1.

## 5 Learning the Representations in Practice

The bounds presented in the previous section suggest that a good representation $\phi$ should be learned to minimize $J_R, J_T$ in (8) and (9). To this end we propose to learn $\phi$ in conjunction with auxiliary models of reward $\mathcal{R}_Z$ and dynamics $\mathcal{P}_Z$. The offline representation learning objective is given by,

$$J_{\mathrm{rep}}(\mathcal{R}_Z, \mathcal{P}_Z, \phi) := \frac{1}{2}\mathbb{E}_{(s,a,r,s')\sim d^{\mathrm{off}}}[\alpha_R \cdot (r - \mathcal{R}_Z(\phi(s), a))^2 - \alpha_T \cdot \log \mathcal{P}_Z(s'|\phi(s), a)], \quad (16)$$

where $\alpha_R, \alpha_T$ are appropriately chosen hyperparameters; in our implementation we choose $\alpha_R = 1, \alpha_T = (1 - \gamma)^{-1}$ to roughly match the coefficients of the bounds in Theorems 2 and 3. Once one chooses parameterizations of $\mathcal{R}_Z, \mathcal{P}_Z$, this objective may be optimized using any stochastic sample-based solver, SGD.

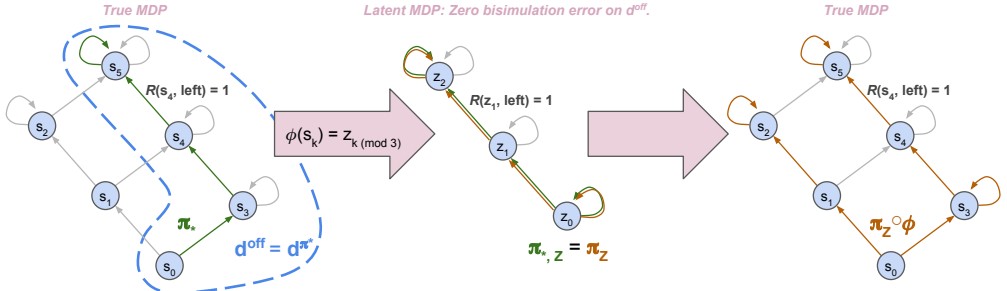

Figure 1: **An example where a latent space model (bisimulation) approach would fail**. The MDP has six states and two actions ('left' and 'right'); arrows denote action dynamics and the green-colored arrows denote the action selection of $\pi_*$ (left); rewards are zero everywhere except for $\mathcal{R}(s_4, \text{left}) = 1$. In this example, we consider an offline distribution $d^{\text{off}} = d^{\pi_*}$, thus there is no distribution shift. The representation $\phi$ given by $\phi(s_k) = z_{k \bmod 3}$ perfectly preserves rewards $(\phi(s), a) \rightarrow r$ and latent transitions $(\phi(s), a) \rightarrow \phi(s')$ on $(s, a) \sim (d^{\text{off}}, \text{Unif}_A)$, ensuring zero bisimulation error. Imitation learning on this representation yields a policy $\pi_Z$ which exactly matches $\pi_{*,Z}$ (orange-colored arrows in the middle) but which achieves significantly worse performance compared to $\pi_*$ on the original MDP (right). Unlike latent space model approaches, our approach measures the dynamics error on $(\phi(s), a) \rightarrow s'$, and so would rightfully reject the $\phi$ presented here.

## 5.1 Contrastive Learning

One may recover a contrastive learning objective by parameterizing $\mathcal{P}_Z$ as an energy-based model. Namely, consider parameterizing $\mathcal{P}_Z$ as

$$\mathcal{P}_Z(s'|z, a) \propto \rho(s') \exp\left\{-||z - g(s', a)||^2/2\right\}, \tag{17}$$

where $\rho$ is a fixed (untrainable) distribution over $S$ (typically set to the distribution of $s'$ in $d^{\text{off}}$) and $g$ is a learnable function $S \times A \rightarrow Z$ (, a neural network). Then $\mathbb{E}_{d^{\text{off}}}[-\log \mathcal{P}_Z(s'|\phi(s), a)]$ yields a contrastive loss:

$$\mathbb{E}_{d^{\text{off}}}[-\log \mathcal{P}_Z(s'|\phi(s), a)] = \frac{1}{2}\mathbb{E}_{d^{\text{off}}}[||\phi(s) - g(s', a)||^2] + \log \mathbb{E}_{\tilde{s}' \sim \rho}[\exp\{-||\phi(s) - g(\tilde{s}', a)||^2/2\}].$$

Similar contrastive learning objectives have appeared in related works [10, 40], and so our theoretical bounds can be used to explain these previous empirical successes.

## 5.2 Linear Models with Contrastive Fourier Features

While the connection between temporal contrastive learning and approximate dynamics models has appeared in previous works [30], it is not immediately clear how one should learn the approximate *linear* dynamics required by Theorem 3. In this section, we show how the same contrastive learning objective can be used to learn approximate linear models, thus illuminating a new connection between contrastive learning and near-optimal sequential decision making; see Appendix A for pseudocode.

We propose to learn a dynamics model $\overline{\mathcal{P}}(s'|s, a) \propto \rho(s') \exp\{-||f(s) - g(s', a)||^2/2\}$ for some functions $f, g$ (, neural networks), which admits a similar contrastive learning objective as mentioned above. Note that this parameterization *does not* involve $\phi$. To recover $\phi$, we may leverage random Fourier features from the kernel literature [33]. Namely, for $k$-dimensional vectors $x, y$ we can approximate $\exp\{-||x - y||^2/2\} \approx \frac{2}{d}\varphi(x)^\top \varphi(y)$, where $\varphi(x) := \cos(Wx + b)$ for $W$ a $d \times k$ matrix with entries sampled from a standard Gaussian and $b$ a vector with entries sampled uniformly from $[0, 2\pi]$. We can therefore approximate $\overline{\mathcal{P}}$ as follows, using $E(s, a) := \mathbb{E}_{\tilde{s}' \sim \rho}[\exp\{-||f(s) - g(\tilde{s}', a)||^2/2\}]$:

$$\overline{\mathcal{P}}(s'|s, a) = \frac{\rho(s')}{E(s, a)} \exp\{-||f(s) - g(s', a)||^2/2\} \approx \frac{2\rho(s')}{d \cdot E(s, a)}\varphi(f(s))^\top \varphi(g(s', a)). \tag{18}$$

Finally, we recover $\phi : S \rightarrow \mathbb{R}^{|A|d}, \psi : S \times A \rightarrow \mathbb{R}^{|A|d}$ as

$$\phi(s) := [\varphi(f(s))/E(s, a)]_{a \in A} \;;\quad \psi(s', a) := [1_{a=\tilde{a}} \cdot 2\rho(s')\varphi(s', \tilde{a})/d]_{\tilde{a} \in A}, \tag{19}$$

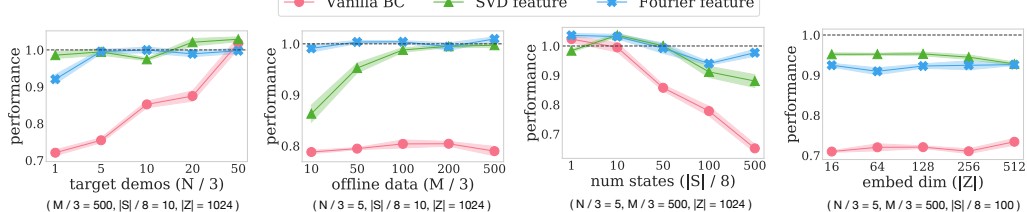

Figure 2: Advantages of representation learning over vanilla behavioral cloning in the tree environment across different $N, M, |S|, |Z|$. Each subplot shows the average performance and standard error across five seeds. Black dotted line shows the performance of the target policy. Representation learning consistently yields significant performance gains.

ensuring $\overline{\mathcal{P}}(s'|s,a) \approx \phi(s)^{\top}\psi(s',a)$ (, $\mathcal{P}_Z(s'|z,a) = \psi(s',a)^{\top}z$), as required for Theorem 3.[3] Notably, during learning explicit computation of $\psi(s',a)$ is never needed, while $E(s,a)$ is straight-forward to estimate from data when $\rho(s') = d^{\text{off}}(s')$,[4] thus making the whole learning procedure – both the offline representation learning and downstream behavioral cloning – practical and easy to implement with deep neural network function approximators for $f, g, \mathcal{R}_Z$; in fact, one can interpret $\phi$ as simply adding an additional untrainable neural network layer on top of $f$, and so these derivations may partially explain why previous works in supervised learning found benefit from adding a non-linear layer on top of representations [15].

## 6 Experiments

We now empirically verify the performance benefits of the proposed representation learning objective in both tabular and Atari game environments. See environment details in Appendix D.

### 6.1 Tree Environments with Low-Rank Transitions

For tabular evaluation, we construct a decision tree environment whose transitions exhibit low-rank structures. To achieve this, we first construct a "canonical" three-level binary tree where each node is associated with a stochastic reward for taking left or right branch and a stochastic transition matrix indicating the probability of landing at either child node. We then duplicate this canonical tree in the state space so that an agent walks down the duplicated tree while the MDP transitions are determined by the canonical tree, thus it is possible to achieve $\epsilon_{\text{R,T}} = 0$ with $|Z| = 8$. We collect the offline data using a uniform random policy and the target demonstrations using a return-optimal policy.

We learn contrastive Fourier features as described in Section 5.2 using tabular $f, g$. We then fix these representations and train a log-linear policy on the target demonstrations. For the baseline, we learn a vanilla BC policy with tabular parametrization directly on target demonstrations.

We also experiment with representations given by singular value decomposition (SVD) of the empirical transition matrix, which is another form of learning factored linear dynamics. Figure 2 shows the performance achieved by the learned policy with and without representation learning. Representation learning consistently yields significant performance gains, especially with few target demonstrations. SVD performs similar to contrastive Fourier features when the offline data is abundant with respect to the state space size, but degrades as the offline data size reduces or the state space grows.

### 6.2 Atari 2600 with Deep Neural Networks

We now study the practical benefit of the proposed contrastive learning objective to imitation learning on 60 Atari 2600 games [11], taking for the offline dataset the DQN Replay Dataset [7], which for

---

[3]While we use random Fourier features with the radial basis kernel, it is clear that a similar derivation can be used with other approximate featurization schemes and other kernels.

[4]In our experiments, we ignore the scaling $E(s,a)$ altogether and simply set $\phi(s) := \varphi(f(s))$. We found this simplification to have little effect on results.

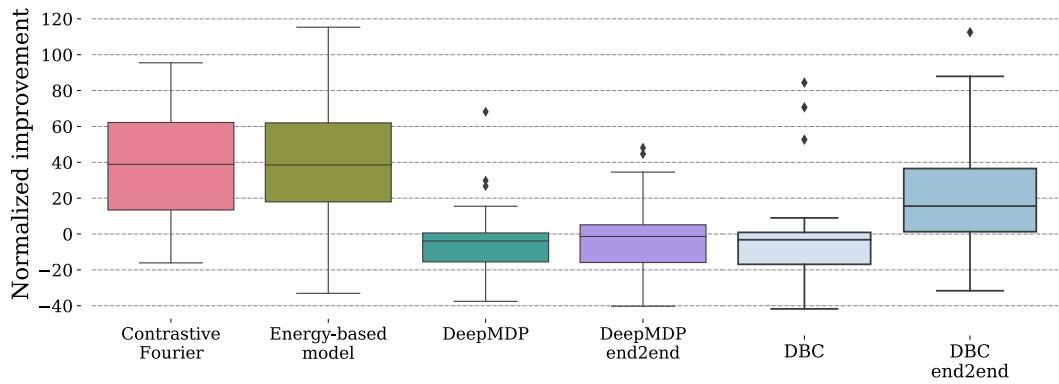

Figure 3: Performance improvements of contrastive Fourier features (setting of Theorem 3), energy-based model (setting of Theorem 2), and bisimulation – DeepMDP [20] and DBC [41] – over vanilla BC. In the baselines, 'end2end' refers to allowing gradients to pass into the representation during downstream imitation learning; by default, the representation is fixed during this phase. Each box and whisker shows the percentiles of normalized improvements (see details in Appendix D) among the 60 Atari games.

each game provides 50M steps collected during DQN training. For the target demonstrations, we take 10k single-step transitions from the last 1M steps of each dataset, corresponding to the data collected near the end of the DQN training. For the offline data, we use all 50M transitions of each dataset.

For our learning agents, we extend the implementations found in Dopamine [14]. We use the standard Atari CNN architecture to embed image-based inputs to vectors of dimension 256. In the case of vanilla BC, we pass this embedding to a log-linear policy and learn the whole network end-to-end with behavioral cloning. For contrastive Fourier features, we use separate CNNs to parameterize $g, f$ in the objective in Section 5.2, and the representation $\phi := \varphi \circ f$ is given by the random Fourier feature procedure described in Section 5.2. A log-linear policy is then trained on top of this representation, but without passing any BC gradients through $\phi$. This corresponds to the setting of Theorem 3. We also experiment with the setting of Theorem 2; in this case the setup is same as for contrastive Fourier features, only that we define $\phi := f$ (, $P_Z$ is an energy-based dynamics model) and we parameterize $\pi_Z$ as a more expressive single-hidden-layer softmax policy on top of $\phi$.

We compare contrastive learning with Fourier features and energy-based models to two latent space models, DeepMDP [20] and Deep Bisimulation for Control (DBC) [41], in Figure 3. Both linear (Fourier features) and energy-based parametrization of contrastive learning achieve dramatic performance gains ($> 40\%$) on over half of the games. DeepMDP and DBC, on the other hand, achieve little improvement over vanilla BC when presented as a separate loss from behavioral cloning. Enabling end-to-end learning of the latent space models as an auxiliary loss to behavioral cloning leads to better performance, but DeepMDP and DBC still underperform contrastive learning. See Appendix D for further ablations.

## 7 Conclusion

We have derived an offline representation learning objective which, when combined with BC, provably minimizes an upper bound on the performance difference from the target policy. We further showed that the proposed objective can be implemented as contrastive learning with an optional projection to Fourier features. Interesting avenues for future work include (1) extending our theory to multi-step contrastive learning, popular in practice [40, 10], (2) deriving similar results for policy learning in offline and online RL settings, and (3) reducing the effect of offline distribution shifts. We also note that our use of contrastive Fourier features for learning a linear dynamics model may be of independent interest, especially considering that a number of theoretical RL works rely on such an approximation (, [4, 39]), while to our knowledge no previous work has demonstrated a practical

and scalable learning algorithm for linear dynamics approximation. Determining if the technique of learning contrastive Fourier features works well for these settings offers another interesting direction to explore.

**Limitations**  One of the main limitations inherent in our theoretical derivations is the dependence on distribution shift, in the form of $1 + D_{\chi^2}(d^{\text{off}} \| d^{\pi_*})^{\frac{1}{2}}$ in all bounds. Arguably, some dependence on the offline distribution is unavoidable: Certainly, if a specific state does not appear in the offline distribution, then there is no way to learn a good representation of it (without further assumptions on the MDP). In practice one potential remedy is to make sure that the offline dataset sufficiently covers the whole state space; the inequality $1 + D_{\chi^2}(p\|q) \leq \|p/q\|_\infty^2$ will ensure this limits the dependence on distribution shift.

## Acknowledgments and Disclosure of Funding

We thank Bo Dai, Rishabh Agarwal, Mohammad Norouzi, Pablo Castro, Marlos Machado, Marc Bellemare, and the rest of the Google Brain team for fruitful discussions and valuable feedback.

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
