# A   Pseudocode

We present basic pseudocode of feature learning below.

---

**Algorithm 1** Representation learning with contrastive energy-based models

---

**Require:** Dataset $\mathcal{D}^{\text{off}}$, parameterized functions $\phi : S \to Z$, $g : S \times A \to \mathbb{R}^d$, $h : Z \times A \to \mathbb{R}$, batch size $B$, weights $\alpha_{\text{R}}, \alpha_{\text{T}}$.
1: **for** $k = 0, 1, 2, \ldots$ **do**
2: Sample $\{(s_i, a_i, r_i, s_i')\}_{i=1}^B \sim \mathcal{D}^{\text{off}}$.state visi
3: Compute reward loss $\ell_{R,i} = (r_i - h(\phi(s_i), a_i))^2$.
4: Compute dynamics loss
  $\ell_{T,i} = \|\phi(s_i) - g(s_i', a_i)\|^2/2 + \log \sum_{j=1}^n \exp\{-\|\phi(s_i) - g(s_j', a_i)\|^2/2\}$.
5: Update $\phi, g, h$ according to loss $\sum_i (\alpha_{\text{R}} \cdot \ell_{R,i} + \alpha_{\text{T}} \cdot \ell_{T,i})$.
6: **return** $\phi$

---

For the Fourier feature representation, we normalize the components of $f$, which, when the inputs $f(s)$ are normally distributed with some unknown mean and variance, may be interpreted as focusing the sampling distribution of $W$ on the most informative Fourier features; mathematically, one may show this still approximates the kernel as $d \to \infty$ by using importance sampling with importance weights placed on $\psi$ instead of $\phi$.

---

**Algorithm 2** Representation learning with contrastive Fourier features

---

**Require:** Dataset $\mathcal{D}^{\text{off}}$, parameterized functions $f : S \to \mathbb{R}^k$, $g : S \times A \to \mathbb{R}^k$, representation dimension $d$, batch size $B$, weights $\alpha_{\text{R}}, \alpha_{\text{T}}$.
1: Set $W \in \mathbb{R}^{d \times k}$ as $W_{i,j} \sim \text{Normal}(0, 1)$.
2: Set $b \in \mathbb{R}^d$ as $b_i \sim \text{Unif}(0, 2\pi)$.
3: Initialize $f_{\text{avg}} = 0, f_{\text{sq}} = 1$.
4: Define normalize as $\text{normalize}(x) = (x - f_{\text{avg}})/\sqrt{f_{\text{sq}}^2 - f_{\text{avg}}^2}$.
5: Define $\phi$ as $\phi(s) = \cos(W \cdot \text{normalize}(f(s)) + b)$.
6: Initialize $h \in \mathbb{R}^{k \times |A|}$ as $h = 0$.
7: **for** $k = 0, 1, 2, \ldots$ **do**
8: Sample $\{(s_i, a_i, r_i, s_i')\}_{i=1}^B \sim \mathcal{D}^{\text{off}}$.
9: Update $f_{\text{avg}}$ with $\frac{1}{B} \sum_i f(s_i)$.
10: Update $f_{\text{sq}}$ with $\frac{1}{B} \sum_i f(s_i)^2$.
11: Compute $z_i = \phi(s_i)$.
12: Compute reward loss $\ell_{R,i} = (r_i - h_{a_i}^\top z_i)^2$.
13: Compute dynamics loss
  $\ell_{T,i} = \|f(s_i) - g(s_i', a_i)\|^2/2 + \log \sum_{j=1}^n \exp\{-\|f(s_i) - g(s_j', a_i)\|^2/2\}$.
14: Update $f, g, h$ according to loss $\sum_i (\alpha_{\text{R}} \cdot \ell_{R,i} + \alpha_{\text{T}} \cdot \ell_{T,i})$.
15: **return** $\phi$

---

# B Proofs

## B.1 Foundational Lemmas

We first present a basic performance difference lemma:

**Lemma 5.** *If $\pi_1$ and $\pi_2$ are two policies in $\mathcal{M}$, then*

$$\text{PerfDiff}(\pi_2, \pi_1) \leq \frac{1}{1 - \gamma} \text{Err}_{d^{\pi_1}}(\pi_1, \pi_2, \mathcal{R}) + \frac{\gamma R_{\max}}{(1 - \gamma)^2} \text{Err}_{d^{\pi_1}}(\pi_1, \pi_2, \mathcal{P}), \qquad (20)$$

*where*

$$\text{Err}_{d^{\pi_1}}(\pi_1, \pi_2, \mathcal{R}) := \left| \mathbb{E}_{s \sim d^{\pi_1}, a_1 \sim \pi_1(s), a_2 \sim \pi_2(s)}[\mathcal{R}(s, a_1) - \mathcal{R}(s, a_2)] \right| \qquad (21)$$

$$\text{Err}_{d^{\pi_1}}(\pi_1, \pi_2, \mathcal{P}) := \sum_{s' \in S} \left| \mathbb{E}_{s \sim d^{\pi_1}, a_1 \sim \pi_1(s), a_2 \sim \pi_2(s)}[\mathcal{P}(s'|s, a_1) - \mathcal{P}(s'|s, a_2)] \right|. \qquad (22)$$

*Note the second quantity above is a scaled TV-divergence between $\mathcal{P} \circ \pi_1 \circ d^{\pi_1}$ and $\mathcal{P} \circ \pi_2 \circ d^{\pi_1}$. When the MDP reward is action-independent, $\mathcal{R}(s, a_1) = \mathcal{R}(s, a_2)$ for all $s \in S, a_1, a_2 \in A$, the same bound holds with the reward term removed.*

*Proof.* Following similar derivations in [2, 30], we express the performance difference in linear operator notation:

$$\text{PerfDiff}(\pi_2, \pi_1) = |\mathcal{R}\Pi_2(I - \gamma\mathcal{P}\Pi_2)^{-1}\mu - \mathcal{R}\Pi_1(I - \gamma\mathcal{P}\Pi_1)^{-1}\mu|, \qquad (23)$$

where $\Pi_1, \Pi_2$ are linear operators $S \to S \times A$ such that $\Pi_i \nu(s, a) = \pi_i(a|s)\nu(s)$. Notice that $d^{\pi_1}$ may be expressed in this notation as $(1 - \gamma)(I - \gamma\mathcal{P}\Pi_1)^{-1}\mu$. We split the expression above into two parts:

$$\text{PerfDiff}(\pi_2, \pi_1) \leq |\mathcal{R}\Pi_2(I - \gamma\mathcal{P}\Pi_2)^{-1}\mu - \mathcal{R}\Pi_2(I - \gamma\mathcal{P}\Pi_1)^{-1}\mu|$$
$$+ |\mathcal{R}\Pi_2(I - \gamma\mathcal{P}\Pi_1)^{-1}\mu - \mathcal{R}\Pi_1(I - \gamma\mathcal{P}\Pi_1)^{-1}\mu|. \qquad (24)$$

We may write the first term above as

$$|\mathcal{R}\Pi_2(I - \gamma\mathcal{P}\Pi_2)^{-1}((I - \gamma\mathcal{P}\Pi_1) - (I - \gamma\mathcal{P}\Pi_2))(I - \gamma\mathcal{P}\Pi_1)^{-1}\mu|$$
$$= (1 - \gamma)^{-1}\gamma \cdot |\mathcal{R}\Pi_2(I - \gamma\mathcal{P}\Pi_2)^{-1}(\mathcal{P}\Pi_2 - \mathcal{P}\Pi_1)d^{\pi_1}|. \qquad (25)$$

Using matrix norm inequalities, we bound the above by

$$(1 - \gamma)^{-1}\gamma \cdot \|\mathcal{R}\Pi_2\|_\infty \|(I - \gamma\mathcal{P}\Pi_2)^{-1}\|_{1,\infty} \cdot |(\mathcal{P}\Pi_2 - \mathcal{P}\Pi_1)d^{\pi_1}|. \qquad (26)$$

We know $\|\mathcal{R}\Pi_2\|_\infty \leq R_{\max}$ and, since $\mathcal{P}\Pi_2$ is a stochastic matrix, $\|(I - \gamma\mathcal{P}\Pi_2)^{-1}\|_{1,\infty} \leq \sum_{t=0}^\infty \gamma^t \|\mathcal{P}\Pi_2\|_{1,\infty} = (1 - \gamma)^{-1}$. Thus, we bound the first term of (24) by

$$\frac{\gamma R_{\max}}{(1 - \gamma)^2}|(\mathcal{P}\Pi_2 - \mathcal{P}\Pi_1)d^{\pi_1}| = \frac{\gamma R_{\max}}{(1 - \gamma)^2}\text{Err}_{d^{\pi_1}}(\pi_1, \pi_2, \mathcal{P}). \qquad (27)$$

For the second term of (24), we have

$$|\mathcal{R}\Pi_2(I - \gamma\mathcal{P}\Pi_1)^{-1}\mu - \mathcal{R}\Pi_1(I - \gamma\mathcal{P}\Pi_1)^{-1}\mu| = (1 - \gamma)^{-1} \cdot |(\mathcal{R}\Pi_2 - \mathcal{R}\Pi_1)d^{\pi_1}| \qquad (28)$$

$$= \frac{1}{1 - \gamma}\text{Err}_{d^{\pi_1}}(\pi_1, \pi_2, \mathcal{R}), \qquad (29)$$

and so we immediately achieve the desired bound in (20).

In the case of action-independent rewards, one may follow the same derivation for the first part of (24), starting with

$$\text{PerfDiff}(\pi_2, \pi_1) = |\mathcal{R}(I - \gamma\mathcal{P}\Pi_2)^{-1}\mu - \mathcal{R}(I - \gamma\mathcal{P}\Pi_1)^{-1}\mu|. \qquad (30)$$

$\square$

We now introduce transition and reward models to proxy $\text{Err}_{d^{\pi_1}}(\pi_1, \pi_2, \mathcal{R})$ and $\text{Err}_{d^{\pi_1}}(\pi_1, \pi_2, \mathcal{P})$:

**Lemma 6.** *For $\pi_1$ and $\pi_2$ two policies in $\mathcal{M}$ and any models $\overline{\mathcal{R}} : S \times A \to [-R_{\max}, R_{\max}], \overline{\mathcal{P}} :$*
*$S \times A \to \Delta(S)$ we have,*

$$\mathrm{Err}_{d^{\pi_1}}(\pi_1, \pi_2, \mathcal{R}) \leq |A|\mathbb{E}_{(s,a) \sim (d^{\pi_1}, \mathrm{Unif}_A)}[|\mathcal{R}(s,a) - \overline{\mathcal{R}}(s,a)|] + \mathrm{Err}_{d^{\pi_1}}(\pi_1, \pi_2, \overline{\mathcal{R}}), \qquad (31)$$

$$\mathrm{Err}_{d^{\pi_1}}(\pi_1, \pi_2, \mathcal{P}) \leq 2|A|\mathbb{E}_{(s,a) \sim (d^{\pi_1}, \mathrm{Unif}_A)}[D_{\mathrm{TV}}(\mathcal{P}(s,a)\|\overline{\mathcal{P}}(s,a))] + \mathrm{Err}_{d^{\pi_1}}(\pi_1, \pi_2, \overline{\mathcal{P}}). \quad (32)$$

*Proof.* For the first bound we have,

$$\mathrm{Err}_{d^{\pi_1}}(\pi_1, \pi_2, \mathcal{R}) = \left|\mathbb{E}_{s \sim d^{\pi_1}, a_1 \sim \pi_1(s), a_2 \sim \pi_2(s)}[\mathcal{R}(s, a_1) - \mathcal{R}(s, a_2)]\right| \qquad (33)$$

$$= \left|\sum_{a \in A} \mathbb{E}_{s \sim d^{\pi_1}}[\mathcal{R}(s,a)\pi_1(a|s) - \mathcal{R}(s,a)\pi_2(a|s)]\right| \qquad (34)$$

$$= \left|\sum_{a \in A} \mathbb{E}_{s \sim d^{\pi_1}}[(\mathcal{R}(s,a) - \overline{\mathcal{R}}(s,a))(\pi_1(a|s) - \pi_2(a|s)) + \overline{\mathcal{R}}(s,a)(\pi_1(a|s) - \pi_2(a|s))]\right| \quad (35)$$

$$\leq \left|\sum_{a \in A} \mathbb{E}_{s \sim d^{\pi_1}}[(\mathcal{R}(s,a) - \overline{\mathcal{R}}(s,a))(\pi_1(a|s) - \pi_2(a|s))]\right| + \mathrm{Err}_{d^{\pi_1}}(\pi_1, \pi_2, \overline{\mathcal{R}}) \qquad (36)$$

$$\leq \sum_{a \in A} \mathbb{E}_{s \sim d^{\pi_1}}[|(\mathcal{R}(s,a) - \overline{\mathcal{R}}(s,a))(\pi_1(a|s) - \pi_2(a|s))|] + \mathrm{Err}_{d^{\pi_1}}(\pi_1, \pi_2, \overline{\mathcal{R}}) \qquad (37)$$

$$\leq |A|\mathbb{E}_{(s,a) \sim (d^{\pi_1}, \mathrm{Unif}_A)}[|\mathcal{R}(s,a) - \overline{\mathcal{R}}(s,a)|] + \mathrm{Err}_{d^{\pi_1}}(\pi_1, \pi_2, \overline{\mathcal{R}}), \qquad (38)$$

as desired. The bound for $\mathcal{P}$ may be similarly derived, noting that $D_{\mathrm{TV}}(\mathcal{P}(s,a)\|\overline{\mathcal{P}}(s,a)) = \frac{1}{2}\sum_{s' \in S} |\mathcal{P}(s'|s,a) - \overline{\mathcal{P}}(s'|s,a)|$. $\qquad\square$

Now we incorporate a representation function $\phi : S \to Z$, showing how the errors above may be further reduced in the special case of $\overline{\mathcal{R}}(s,a) = \mathcal{R}_Z(\phi(s),a), \overline{\mathcal{P}}(s,a) = \mathcal{P}_Z(\phi(s),a)$:

**Lemma 7.** *Let $\phi : S \to Z$ for some space $Z$ and suppose there exist $\mathcal{R}_Z : Z \times A \to [-R_{\max}, R_{\max}]$ and $\mathcal{P}_Z : Z \times A \to \Delta(S)$ such that $\overline{\mathcal{R}}(s,a) = \mathcal{R}_Z(\phi(s),a)$ and $\overline{\mathcal{P}}(s,a) = \mathcal{P}_Z(\phi(s),a)$ for all $s \in S, a \in A$. Then for any policies $\pi_1, \pi_2$,*

$$\mathrm{Err}_{d^{\pi_1}}(\pi_1, \pi_2, \overline{\mathcal{R}}) \leq 2R_{\max}\mathbb{E}_{z \sim d_Z^{\pi_1}}[D_{\mathrm{TV}}(\pi_{1,Z}\|\pi_{2,Z})], \qquad (39)$$

$$\mathrm{Err}_{d^{\pi_1}}(\pi_1, \pi_2, \overline{\mathcal{P}})] \leq 2\mathbb{E}_{z \sim d_Z^{\pi_1}}[D_{\mathrm{TV}}(\pi_{1,Z}\|\pi_{2,Z})], \qquad (40)$$

*where,*

$$d_Z^{\pi_1}(z) := \mathrm{Pr}[z = \phi(s) \mid s \sim d^{\pi_1}], \qquad (41)$$

$$\pi_{k,Z}(z) := \mathbb{E}[\pi_k(s) \mid s \sim d^{\pi_1}, z = \phi(s)], \qquad (42)$$

*for all $z \in Z, k \in \{1, 2\}$.*

*Proof.* The result follows from straightforward algebraic manipulation via the definitions of $d_Z^{\pi_1}, \pi_{k,Z}$ and triangle inequality. For the reward error, we have,

$$\left| \mathbb{E}_{s \sim d^{\pi_1}, a_1 \sim \pi_1(s), a_2 \sim \pi_2(s)} [\overline{\mathcal{R}}(s, a_1) - \overline{\mathcal{R}}(s, a_2)] \right| \tag{43}$$

$$= \left| \sum_{s \in S, a \in A} \mathcal{R}_Z(\phi(s), a) \pi_1(a|s) d^{\pi_1}(s) - \sum_{s \in S, a \in A} \mathcal{R}_Z(\phi(s), a) \pi_2(a|s) d^{\pi_1}(s) \right|$$

$$= \left| \sum_{z \in Z, a \in A} \mathcal{R}_Z(z, a) \sum_{s \in S, \phi(s) = z} \pi_1(a|s) d^{\pi_1}(s) - \sum_{z \in Z, a \in A} \mathcal{R}_Z(z, a) \sum_{s \in S, \phi(s) = z} \pi_2(a|s) d^{\pi_1}(s) \right|$$

$$= \left| \sum_{z \in Z, a \in A} \mathcal{R}_Z(z, a) \pi_{1,Z}(a|z) d_Z^{\pi_1}(z) - \sum_{z \in Z, a \in A} \mathcal{R}_Z(z, a) \pi_{2,Z}(a|z) d_Z^{\pi_1}(z) \right|$$

$$= \left| \mathbb{E}_{z \sim d_Z^{\pi_1}} \left[ \sum_{a \in A} \mathcal{R}_Z(z, a) (\pi_{1,Z}(a|z) - \pi_{2,Z}(a|z)) \right] \right| \tag{44}$$

$$\leq \mathbb{E}_{z \sim d_Z^{\pi_1}} \left[ \sum_{a \in A} |\mathcal{R}_Z(z, a) (\pi_{1,Z}(a|z) - \pi_{2,Z}(a|z))| \right] \tag{45}$$

$$\leq R_{\max} \mathbb{E}_{z \sim d_Z^{\pi_1}} \left[ \sum_{a \in A} |\pi_{1,Z}(a|z) - \pi_{2,Z}(a|z)| \right] = 2 R_{\max} \mathbb{E}_{z \sim d_Z^{\pi_1}} \left[ D_{\mathrm{TV}}(\pi_{1,Z} \| \pi_{2,Z}) \right], \tag{46}$$

as desired. The bound for the transition error may be derived analogously. $\square$

In the case of *linear* reward and transition models, we may derive a variant of Lemma 7, which will be useful in the proof of Theorem 3:

**Lemma 8.** *Let* $\phi : S \to Z$ *for some* $Z \subset \mathbb{R}^d$ *and suppose there exist* $r : A \to \mathbb{R}^d$ *and* $\psi : S \times A \to \mathbb{R}^d$ *such that* $\overline{\mathcal{R}}(s, a) = r(a)^\top \phi(s)$ *and* $\overline{\mathcal{P}}(s, a) = \psi(s', a)^\top \phi(s)$ *for all* $s, s' \in S, a \in A$. *Let* $\pi : S \to \Delta(A)$ *be any policy in* $\mathcal{M}$ *and* $\pi_\theta(z) := \mathrm{softmax}(\theta^\top z)$ *be a latent policy for some* $\theta \in \mathbb{R}^{d \times |A|}$. *Then,*

$$\mathrm{Err}_{d^\pi}(\pi, \pi_\theta, \overline{\mathcal{R}}) \leq \|r\|_\infty \cdot \left\| \frac{\partial}{\partial \theta} \mathbb{E}_{s \sim d^\pi, a \sim \pi(s)} [- \log \pi_\theta(a|\phi(s))] \right\|_1, \tag{47}$$

$$\mathrm{Err}_{d^\pi}(\pi, \pi_\theta, \overline{\mathcal{P}}) \leq \|\psi\|_\infty \cdot \left\| \frac{\partial}{\partial \theta} \mathbb{E}_{s \sim d^\pi, a \sim \pi(s)} [- \log \pi_\theta(a|\phi(s))] \right\|_1, \tag{48}$$

*where* $\|r\|_\infty = \max_{a \in A} \|r(a)\|_\infty$ *and* $\|\psi\|_\infty = \max_{s' \in S, a \in A} \|\psi(s', a)\|_\infty$.

*Proof.* The crux of the proof is noting that the gradient above for a specific column $\theta_a$ of $\theta$ may be expressed as

$$\frac{\partial}{\partial \theta_a} \mathbb{E}_{s \sim d^\pi, \tilde{a} \sim \pi(s)} [- \log \pi_\theta(\tilde{a}|\phi(s))] = \frac{\partial}{\partial \theta_a} \left( \mathbb{E}_{s \sim d^\pi, \tilde{a} \sim \pi(s)} [- \theta_{\tilde{a}}^\top \phi(s)] + \mathbb{E}_{s \sim d^\pi} [\log \sum_{\tilde{a} \in A} \exp\{\theta_{\tilde{a}}^\top \phi(s)\}] \right)$$

$$= - \mathbb{E}_{s \sim d^\pi} [\phi(s) \pi(a|s)] + \mathbb{E}_{s \sim d^\pi} [\phi(s) \pi_\theta(a|\phi(s))], \tag{49}$$

and so,

$$r(a)^\top \frac{\partial}{\partial \theta_a} \mathbb{E}_{s \sim d^\pi, a \sim \pi(s)} [- \log \pi_\theta(a|\phi(s))]$$

$$= - \mathbb{E}_{s \sim d^\pi} [\pi(a|s) \cdot \overline{\mathcal{R}}(s, a)] + \mathbb{E}_{s \sim d^\pi} [\pi_\theta(a|\phi(s)) \cdot \overline{\mathcal{R}}(s, a)]. \tag{50}$$

Summing over $a \in A$, we have,

$$\sum_{a \in A} r(a)^\top \frac{\partial}{\partial \theta_a} \mathbb{E}_{s \sim d^\pi, a \sim \pi(s)} [- \log \pi_\theta(a|\phi(s))] = \mathbb{E}_{s \sim d^\pi, a_1 \sim \pi(s), a_2 \sim \pi_\theta(s)} [- \overline{\mathcal{R}}(s, a_1) + \overline{\mathcal{R}}(s, a_2)].$$

Thus, we have,

$$\text{Err}_{d^\pi}(\pi, \pi_\theta, \overline{\mathcal{R}}) = \left| \mathbb{E}_{s \sim d^\pi, a_1 \sim \pi(s), a_2 \sim \pi_\theta(s)}[-\overline{\mathcal{R}}(s, a_1) + \overline{\mathcal{R}}(s, a_2)] \right| \tag{51}$$

$$= \left| \sum_{a \in A} r(a)^\top \frac{\partial}{\partial \theta_a} \mathbb{E}_{s \sim d^\pi, \tilde{a} \sim \pi(s)}[-\log \pi_\theta(\tilde{a}|\phi(s))] \right| \tag{52}$$

$$\leq \sum_{a \in A} \|r(a)\|_\infty \cdot \left\| \frac{\partial}{\partial \theta_a} \mathbb{E}_{s \sim d^\pi, \tilde{a} \sim \pi(s)}[-\log \pi_\theta(\tilde{a}|\phi(s))] \right\|_1 \tag{53}$$

$$\leq \|r\|_\infty \cdot \left\| \frac{\partial}{\partial \theta} \mathbb{E}_{s \sim d^\pi, a \sim \pi(s)}[-\log \pi_\theta(a|\phi(s))] \right\|_1, \tag{54}$$

as desired. The bound for the transition error may be derived analogously. $\square$

Our final lemma will be used to translate on-policy bounds to off-policy.

**Lemma 9.** *For two distributions $\rho_1, \rho_2 \in \Delta(S)$ with $\rho_1(s) > 0 \Rightarrow \rho_2(s) > 0$, we have,*

$$\mathbb{E}_{\rho_1}[h(s)] \leq (1 + D_{\chi^2}(\rho_1 \| \rho_2)^{\frac{1}{2}}) \sqrt{\mathbb{E}_{\rho_2}[h(s)^2]}. \tag{55}$$

*Proof.* The lemma is a straightforward consequence of Cauchy-Schwartz:

$$\mathbb{E}_{\rho_1}[h(s)] = \mathbb{E}_{\rho_2}[h(s)] + (\mathbb{E}_{\rho_1}[h(s)] - \mathbb{E}_{\rho_2}[h(s)]) \tag{56}$$

$$= \mathbb{E}_{\rho_2}[h(s)] + \sum_{s \in S} \frac{\rho_1(s) - \rho_2(s)}{\rho_2(s)^{\frac{1}{2}}} \cdot \rho_2(s)^{\frac{1}{2}} h(s) \tag{57}$$

$$\leq \mathbb{E}_{\rho_2}[h(s)] + \left( \sum_{s \in S} \frac{(\rho_1(s) - \rho_2(s))^2}{\rho_2(s)} \right)^{\frac{1}{2}} \cdot \left( \sum_{s \in S} \rho_2(s) h(s)^2 \right)^{\frac{1}{2}} \tag{58}$$

$$= \mathbb{E}_{\rho_2}[h(s)] + D_{\chi^2}(\rho_1 \| \rho_2)^{\frac{1}{2}} \cdot \sqrt{\mathbb{E}_{\rho_2}[h(s)^2]}. \tag{59}$$

Finally, to get the desired bound, we simply note that the concavity of the square-root function implies $\mathbb{E}_{\rho_2}[h(s)] \leq \mathbb{E}_{\rho_2}[\sqrt{h(s)^2}] \leq \sqrt{\mathbb{E}_{\rho_2}[h(s)^2]}$. $\square$

### B.2 Proof of Theorem 2

With the lemmas in the above section, we are now prepared to prove Theorem 2. We begin with the following on-policy version of Theorem 2 and then derive the off-policy bound:

**Lemma 10.** *Consider a representation function $\phi : S \rightarrow Z$ and models $\mathcal{R}_Z : Z \times A \rightarrow [-R_{\max}, R_{\max}], \mathcal{P}_Z : Z \times A \rightarrow \Delta(S)$. Denote the representation error as*

$$\epsilon_{\text{R,T}} := \frac{|A|}{1 - \gamma} \mathbb{E}_{(s,a) \sim (d^{\pi_*}, \text{Unif}_A)}[|\mathcal{R}(s, a) - \mathcal{R}_Z(\phi(s), a)|]$$

$$+ \frac{2\gamma|A|R_{\max}}{(1 - \gamma)^2} \mathbb{E}_{(s,a) \sim (d^{\pi_*}, \text{Unif}_A)}[D_{\text{TV}}(\mathcal{P}(s, a) \| \mathcal{P}_Z(\phi(s), a))]. \tag{60}$$

*Then the performance difference between $\pi_*$ and a latent policy $\pi_Z : Z :\rightarrow \Delta(A)$ may be bounded as,*

$$\text{PerfDiff}(\pi_Z, \pi_*) \leq \epsilon_{\text{R,T}} + C \sqrt{\frac{1}{2} \underbrace{\mathbb{E}_{z \sim d_Z^{\pi_*}}[D_{\text{KL}}(\pi_{*,Z}(z) \| \pi_Z(z))]}_{= \text{const}(\pi_*, \phi) + J_{\text{BC}, \phi}(\pi_Z)}}, \tag{61}$$

*where $C = \frac{2R_{\max}}{(1-\gamma)^2}$ and $d_Z^{\pi_*}, \pi_{*,Z}$ are the marginalization of $d^{\pi_*}, \pi_*$ onto $Z$ according to $\phi$:*

$$d_Z^{\pi_*}(z) := \Pr[z = \phi(s) \mid s \sim d^{\pi_*}]; \quad \pi_{*,Z}(z) := \mathbb{E}[\pi_*(s) \mid s \sim d^{\pi_*}, z = \phi(s)]. \tag{62}$$

*When the MDP reward is action-independent, $\mathcal{R}(s, a_1) = \mathcal{R}(s, a_2)$ for all $s \in S, a_1, a_2 \in A$, the same bound holds with the reward term removed from $\epsilon_{\text{R,T}}$.*

*Proof.* We combine Lemmas 5, 6, and 7 with $\pi_1 := \pi_*$ and $\pi_2 := \pi_Z \circ \phi$ to yield the following bound:

$$\text{PerfDiff}(\pi_Z, \pi_*) \leq \epsilon_{\text{R,T}} + C \cdot \mathbb{E}_{z \sim d_Z^{\pi_*}}[D_{\text{TV}}(\pi_{*,Z}(z) \| \pi_Z(z))]. \tag{63}$$

To yield the desired bound, we simply apply Pinsker's inequality and the concavity of the square-root function:

$$\mathbb{E}_{z \sim d_Z^{\pi_*}}[D_{\text{TV}}(\pi_{*,Z}(z) \| \pi_Z(z))] \leq \mathbb{E}_{z \sim d_Z^{\pi_*}}\left[\sqrt{\frac{1}{2}D_{\text{KL}}(\pi_{*,Z}(z) \| \pi_Z(z))}\right] \tag{64}$$

$$\leq \sqrt{\frac{1}{2}\mathbb{E}_{z \sim d_Z^{\pi_*}}[D_{\text{KL}}(\pi_{*,Z}(z) \| \pi_Z(z))]}. \tag{65}$$

$\square$

The proof of Theorem 2 then immediately follows from application of Lemma 9 to the bound in Lemma 10, noting that (again due to Pinsker's) $D_{\text{TV}}(\mathcal{P}(s,a) \| \mathcal{P}_Z(\phi(s), a))^2 \leq \frac{1}{2}D_{\text{KL}}(\mathcal{P}(s,a) \| \mathcal{P}_Z(\phi(s), a))$

### B.3    Proof of Theorem 3

The proof of Theorem 3 is derived analogously to that of Theorem 2 above, except using Lemma 8 in place of Lemma 7.

### B.4    Proof of Lemma 1

Lemma 1 may be immediately derived from Theorem 2, using $Z := S$ and $\phi(s) := s$ with $\mathcal{R}_Z := \mathcal{R}$ and $\mathcal{P}_Z := \mathcal{P}$.

## C    Sample Efficiency

**Lemma 11.** *Let $\rho \in \Delta(\{1, \ldots, k\})$ be a distribution with finite support. Let $\hat{\rho}_n$ denote the empirical estimate of $\rho$ from $n$ i.i.d. samples $X \sim \rho$. Then,*

$$\mathbb{E}_n[D_{\text{TV}}(\rho \| \hat{\rho}_n)] \leq \frac{1}{2} \cdot \frac{1}{\sqrt{n}} \sum_{i=1}^{k} \sqrt{\rho(i)} \leq \frac{1}{2} \cdot \sqrt{\frac{k}{n}}. \tag{66}$$

*Proof.* The first inequality is Lemma 8 in [12] while the second inequality is due to the concavity of the square root function. $\square$

**Lemma 12.** *Let $\mathcal{D} := \{(s_i, a_i)\}_{i=1}^{n}$ be i.i.d. samples from a factored distribution $x(s, a) := \rho(s)\pi(a|s)$ for $\rho \in \Delta(S), \pi : S \to \Delta(A)$. Let $\hat{\rho}$ be the empirical estimate of $\rho$ in $\mathcal{D}$ and $\hat{\pi}$ be the empirical estimate of $\pi$ in $\mathcal{D}$. Then,*

$$\mathbb{E}_{\mathcal{D}}[\mathbb{E}_{s \sim \rho}[D_{\text{TV}}(\pi(s) \| \hat{\pi}(s))]] \leq \sqrt{\frac{|S||A|}{n}}. \tag{67}$$

*Proof.* Let $\hat{x}$ be the empirical estimate of $x$ in $\mathcal{D}$. We have,

$$\mathbb{E}_{s\sim\rho}[D_{\text{TV}}(\pi(s)\|\hat{\pi}(s))] = \frac{1}{2}\sum_{s,a}\rho(s)\cdot|\pi(a|s) - \hat{\pi}(a|s)| \tag{68}$$

$$= \frac{1}{2}\sum_{s,a}\rho(s)\cdot\left|\frac{x(s,a)}{\rho(s)} - \frac{\hat{x}(s,a)}{\hat{\rho}(s)}\right| \tag{69}$$

$$\leq \frac{1}{2}\sum_{s,a}\rho(s)\cdot\left|\frac{\hat{x}(s,a)}{\rho(s)} - \frac{\hat{x}(s,a)}{\hat{\rho}(s)}\right| + \frac{1}{2}\sum_{s,a}\rho(s)\cdot\left|\frac{\hat{x}(s,a)}{\rho(s)} - \frac{x(s,a)}{\rho(s)}\right| \tag{70}$$

$$= \frac{1}{2}\sum_{s,a}\rho(s)\cdot\left|\frac{\hat{x}(s,a)}{\rho(s)} - \frac{\hat{x}(s,a)}{\hat{\rho}(s)}\right| + D_{\text{TV}}(x\|\hat{x}) \tag{71}$$

$$= \frac{1}{2}\sum_{s}\rho(s)\cdot\left|\frac{1}{\rho(s)} - \frac{1}{\hat{\rho}(s)}\right|\left(\sum_{a}\hat{x}(s,a)\right) + D_{\text{TV}}(x\|\hat{x}) \tag{72}$$

$$= \frac{1}{2}\sum_{s}\rho(s)\cdot\left|\frac{1}{\rho(s)} - \frac{1}{\hat{\rho}(s)}\right|\cdot\hat{\rho}(s) + D_{\text{TV}}(x\|\hat{x}) \tag{73}$$

$$= D_{\text{TV}}(\rho\|\hat{\rho}) + D_{\text{TV}}(x\|\hat{x}). \tag{74}$$

Finally, the bound in the lemma is achieved by application of Lemma 11 to each of the TV divergences. $\square$

### C.1 Proof of Theorem 4

To prove Theorem 4, we first present the following variant of the bound in Theorem 3, which maintains the BC loss as a TV divergence rather than a KL divergence and whose validity is clear from (63):

$$\text{PerfDiff}(\pi_Z, \pi_*) \leq (1 + D_{\chi^2}(d^{\pi_*}\|d^{\text{off}})^{\frac{1}{2}})\cdot\epsilon_{\text{R,T}} + C\cdot\mathbb{E}_{z\sim d_Z^{\pi_*}}[D_{\text{TV}}(\pi_{*,Z}(z)\|\pi_Z(z))], \tag{75}$$

where $C = \frac{2R_{\max}}{(1-\gamma)^2}$.

The result in Theorem 4 is then derived by setting $\pi_Z := \pi_{N,Z}$ and using the result of Lemma 12, noting that learning a tabular $\pi_Z$ with BC on $\mathcal{D}_N^{\pi_*}$ corresponds to setting $\pi_Z(a|z)$ to be the empirical conditional distribution appearing in $\mathcal{D}_N^{\pi_*}$ with respect to $\phi_M$.

## D Experiment Details

### D.1 Tree Environment Details

The three-level binary decision tree used in Section 6 has a $0.8$ chance of landing on the intended child node and a $0.2$ chance of random exploration following the Dirichlet distribution ($\alpha = 1$). Each node in the tree has two rewards associated with taking left or right actions. The mean of the rewards are generated uniformly between $0$ and $1$ and are powered to the third and normalized to have a maximum of $1$ at each step. A unit Gaussian noise is then applied to the rewards. An optimal policy and a random policy achieve near $1$ and $0.5$ average per-step reward respectively.

### D.2 Experiment Details

**Tree** For tree experiments in Figure 2, we set $|S|/8 = 10$, $M/3 = 500$, $|Z| = 1024$ when ablating over $N$, $|S|/8 = 10$, $N/3 = 5$, $|Z| = 1024$ when ablating over $M$, $N/3 = 5$, $M/3 = 500$, $|Z| = 1024$ when ablating over $|S|$, and $N/3 = 5$, $M/3 = 500$, $|S|/8 = 100$ when ablating over $|Z|$. For the representation dimensions, we use 16 SVD features or 1024 Fourier features. We use the Adam optimizer with learning rate $0.01$ for both representation learning and behavioral cloning.

**Atari**   For Atari experiments in Figure 3, we set state embedding size to 256, Fourier features size to 8192 and use all 50M transitions as the offline data by default. When sampling batches, to encourage better negative samples in the contrastive learning, we sample 4 sequences of length 64 from the replay buffer, making a total batch size of 256 single-step transitions. We use the standard Atari CNN architecture with three convolutional layers interleaved with ReLU activation followed by two fully connected layers with 512 units each to output state representations. For the energy-based parametrization in Section D.4, we $\pi_Z$ is a single-hidden-layer NN with 512 units. All networks are trained using the Adam optimizer with learning rate 0.0001. We train the representations and behavior cloning concurrently with separate losses. Training is conducted on NVIDIA P100 GPUs.

### D.3   Improvements on individual Atari games

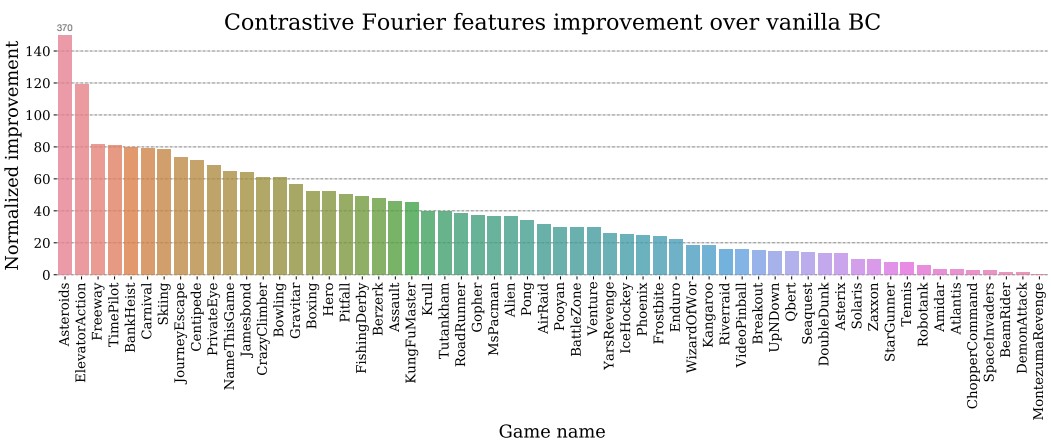

Figure 4: Average performance improvements (over 5 seeds) of contrastive Fourier features over vanilla BC on individual Atari games.

### D.4   Ablation study of contrastive learning

We further investigate various factors that potentially affect the benefit of contrastive representation learning. We consider:

- **Size of the state representations**: Either 256 or 512.
- **Parameterization of the approximate dynamics model**: Either using Fourier features with a log-linear policy (corresponding to Section 5.2 and Theorem 3) or using contrastive learning of an energy-based dynamics model with a more expressive single-hidden-layer neural network for $\pi_Z$ (corresponding to Section 5.1 and Theorem 2).
- **How far the offline distribution differs from target demonstrations**: Using either the last 10M, 25M, or whole 50M transitions in the offline replay dataset.

Figure 5 shows the quantile of the normalized improvements among the 60 Atari games. Overall, the benefit of representation learning is robust to these factors, and is slightly more pronounced with larger state embedding size and smaller difference between the offline distribution and target demonstrations.

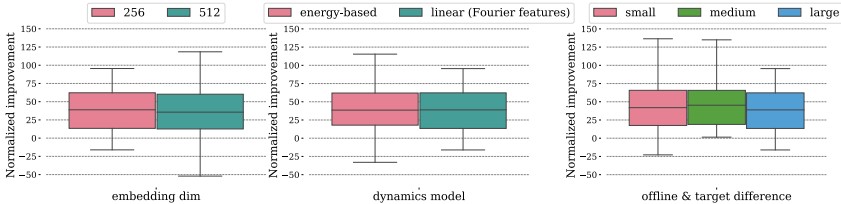

Figure 5: Ablations over embedding dimension, parametrization of the approximate dynamics model, and difference between the offline distribution and target demonstrations. Each ablation changes one factor from the default (see Appendix D), and shows the quantile (among 60 Atari games) of the normalized improvements. Representation learning consistently shows significant gains.