# OpenReview forum: "Provable Representation Learning for Imitation with Contrastive Fourier Features"
_NeurIPS.cc/2021/Conference — NeurIPS 2021 Poster_

### Official Review · Reviewer_pRgN · 2021-07-03

**Rating:** 6
**Confidence:** 3

**Summary:**

This paper proposes an approach for learning a low-dimensional state representation that favors the training procedure of a policy with imitation learning. One of the main challenges is that the representation learning strategy is run on offline data, and is possibly not informed about the expert policy to imitate. The paper presents a theoretical framework to bound the performance of the student policy in function of the expressiveness of the representation and the distance to the distance between expert and student on low-dimensional states. The theoretical findings are then put into practice by deriving an algorithm to learn representations via contrastive learning and Fourier features. The algorithm is then evaluated on both toy tabular examples and on more complex high-dimensional tasks (Atari).

**Limitations And Societal Impact:**

No remark at all was done about the possible negative societal impacts of this work. I think this is acceptable since the contribution is mainly theoretical, but I would have appreciated it if the paper would have at least mentioned it (for example, can such work be used for military applications?).

**Main Review:**

The main findings of this work are, to the best of my knowledge, novel. The paper does a very good job of covering the (large) field of representation learning for control. The most important related work is adequately cited, and some interesting connections are drawn between prior empirical evidence and the proposed method. However, it would have been nice to also mention and discuss existing work on learning representation by maximizing information for the task [1].

I find the paper presentation very good, well organized, and easy to follow (except for some minor comments that I report below). However, I am a bit concerned that very few technical details are given for the key experiments of the paper, i.e., the one on Atari games. Indeed, only there learning a representation for the states (i.e., images) actually makes sense (the other demonstrator is just a toy tabular MDP). No mention is made of the architecture, the learning rate, the training time, etc., which makes the experiment basically not reproducible. I have checked in the supplementary code, but I only found code related to the toy tabular MDP and not of the Atari games.

Regarding the technical quality of the submission, I think that the derivations are sound and supported by empirical evidence. On the negative side, I believe this work's biggest weakness is the lack of a clear discussion about the limitations of the proposed method. More specifically, Eq. 11, which presents the central finding of the paper, shows how the student performance is bound in terms of the representation power and the distance expert-student in representation space. However, it is completely unclear what are, in the general settings, the dominant terms of this bound. My concern is that the state observations probability is pretty different between an expert and offline demonstrations, giving a large  $D_{\chi^2}(d^{\pi} || d^{off})$. Such component will probably also be amplified by the loss of representations $\epsilon_{R, T}$ (which I would doubt to be below one). This probably becomes even worse as the representation and policies are forced to be log-linear. As a result, I feel that the bound is completely dominated by its first term and de facto loose for most cases. Note that such a remark does not diminish the derived practical algorithm, which is interesting and with good performance. However, I think that the paper should be more open about this potential issue and discuss how to address it. On a similar note, it would be nice to clarify the effect of realizability of the expert policy on the low-level representation z (the so-called $const(\pi_*, \phi)$, which could actually also pretty large if the demonstrations and expert policy are very different.

Other minor comments to improve the presentation and strengthen the contribution:

1. Include stronger baselines for representation learning (e.g. [39]  or several other methods mentioned in the related work).
2. After each lemma/theorem report in which part of the appendix the proof can be found.
3. Section 4.3 is titled sample efficiency, but it has nothing to do with the number of samples required to training the policy. It only specializes the bound of Eq.11 to a tabular setting, as far as I understood.
4. It would be nice to clarify that the low-level representation needs to encode enough information to solve the downstream task. Otherwise, some trivial solutions could be found (e.g., putting everything to zero).

Overall, I think this is a nice and interesting paper. The findings could be useful to the control learning community and open new ways to learn policies from high-level observations in a simple and (possibly) provable fashion. However, the contribution could be strongly enhanced by openly discussing the limitations of the theory and algorithm, ensuring reproducibility, and empirically comparing with stronger (or simply more related) baselines.

[1] A Separation Principle for Control in the Age of Deep Learning, Achille, et al.

**Time Spent Reviewing:**

4

---

> ### Author Response · Authors · 2021-08-09
> **Response**
>
> We appreciate the reviewer’s close reading of the paper. We are glad the reviewer found the work novel and technical derivations sound. We address the reviewer’s concerns below. Please let us know if these adequately address all your concerns, or if you have any further questions.
>
> “it would have been nice to also mention and discuss existing work on learning representation by maximizing information for the task [1]”
>
> Thanks for pointing us to this related work. We will include a discussion of it in the final version of the paper.
>
> “No mention is made of the architecture, the learning rate, the training time, etc.”
>
> Many of these details are already included in Appendix D of the supplementary material. We will add further details to ensure reproducibility.
>
> “I think that the paper should be more open about this potential issue and discuss how to address it”
>
> Thanks for the feedback. We will be sure to include more detailed discussions of limitations like too much of an offline distribution shift in the final version of the paper. Arguably, the dependence on the offline distribution is unavoidable to some extent: Certainly, if a specific state doesn’t appear in the offline distribution, then there is no way to learn a good representation of it (without further assumptions on the MDP). Still, in practice one potential remedy is to just make sure that the offline dataset sufficiently covers the whole state space: the inequality $1 + D_{\chi^2}(p\|q) \le \|p/q\|_\infty^2$ will ensure this limits the dependence on distribution shift.
>
> “it would be nice to clarify the effect of realizability of the expert policy on the low-level representation z (the so-called const(π∗,ϕ)), which could actually also pretty large if the demonstrations and expert policy are very different”
>
> This is incorrect. We emphasize: realizability of the expert policy in z is *irrelevant* to the 2nd term of Theorem 2. In fact, there always exists a latent policy which makes the 2nd term exactly zero, regardless of the form of $\pi_*$; this latent policy is simply $\pi_{*,Z}$, as defined in Theorem 2.
>
> “Include stronger baselines for representation learning (e.g. [39] or several other methods mentioned in the related work).”
>
> |  |  Contrastive Fourier  |  DBC  |  DBC end-to-end |
> |------------------|:----:|:----:|:----:|
> | Median          |  39 |   -2  |   18  |
> | 25th percentile | 15 |   -18  | 1 |
> | 75th percentile  | 62 | 0 | 38 |
>
> Table 1. Percentiles of normalized improvements of contrastive Fourier features and deep bisimulation for control (DBC) [39] over vanilla BC among the 60 Atari games.
>
> Thanks for the suggestion. We evaluate [39] in Table 1 following the same setting as Figure 3 in the original paper. While [39] is stronger than DeepMDP,  contrastive Fourier features still outperforms [39] significantly. We will plan to include these additional baselines in the final version of the paper.
>
> “After each lemma/theorem report in which part of the appendix the proof can be found.”
>
> Thanks, this is a good suggestion.
>
> “Section 4.3 is titled sample efficiency, but it has nothing to do with the number of samples required to training the policy. It only specializes the bound of Eq.11 to a tabular setting, as far as I understood.”
>
> The main development in Theorem 4 is the second term (of the RHS) in Eq 15, which establishes a dependence of $N$ (the number of samples in the target demonstrations) on the quality of the final learned policy. Specifically, this bound shows that the dependence on $N$ is less than that without representation learning.
>
> “It would be nice to clarify that the low-level representation needs to encode enough information to solve the downstream task. Otherwise, some trivial solutions could be found (e.g., putting everything to zero).”
>
> This is incorrect. Trivial solutions would necessarily incur a high representation error $\epsilon_{R,T}$. This is due to the fact that $\epsilon_{R,T}$ measures dynamics divergence w.r.t. next *raw* (as opposed to latent) observations; this is the crucial difference from bisimulation principles elaborated in Section 4.4. Accordingly, the need to “encode enough information” is completely encapsulated in the representation error $\epsilon_{R,T}$.

---

> > ### Comment · Reviewer_pRgN · 2021-08-27
> > **Thanks for your answer!**
> >
> > Thanks for clarifying my doubts about the bound. I hope the in future versions of the manuscript, the paper will be open about the possible limitations deriving from the distribution shift. However, I am still not entirely convinced about the latest answer on representation error. If the error is just computed on the basis of raw information, how can we guarantee that the hidden representation contains enough information for the downstream task? Simple counter-example: You want to detect (small) flying vehicles on a colored background. Since the objects of interest are small, only reconstructing the background would result in a small representation error. However, the hidden representation would not have the required information to solve the task. Could the authors comment on this?

---

> > > ### Author Response · Authors · 2021-08-27
> > > **Response**
> > >
> > > Thanks for the updated example, which is more interesting than "putting everything to zero".
> > >
> > > The issue in this example is that although the representation error is "small" (because the flying vehicles are small compared to the background), it is only **relatively** small (relative to a uniform random predictor). In contrast, in absolute terms (and in the context of our bounds) the error will be large. We can think of a simple example to illustrate this. Suppose we are trying to predict a k-length vector of bits, where the first bit is dependent on the action and the remaining bits are always 0. The dynamics portion of the representation loss will be something like $E_{bitvector}[- \log P(bitvector | action)] = E_{bitvector}[-\sum_{i} \log P(bitvector[i] | action) ]$. Due to the sum, it is clear that when the i>=1 bits are perfectly predicted, the total loss is the same as the dynamics loss measured only on the first bit -- which is effectively looking at the same environment but without the "distractor" bits. Thus, since our bounds hold in the first-bit-only, distractor-less environment, they should also hold in the distractor-filled form of the environment.
> > >
> > > Still, this example is a nice situation that highlights the difference between theory and practice. Even if the error appears relatively small in practice, it may still be large in absolute terms (i.e., in terms of theoretical bounds). This example also highlights many important implementation choices: e.g., one advantage of energy-based dynamics models (which we use in our experiments) is that they don't have as much of an issue with these sorts of distractors as compared to more traditional generative dynamics models.

---

> > > > ### Comment · Reviewer_pRgN · 2021-08-30
> > > > **Thanks for the answer.**
> > > >
> > > > Thanks for clarifying this point. I agree with your arguments. However, I think that the paper should be open about this possible issue, and possibly evaluate in a quantitative fashion the statements of the previous answer. But overall, I think this could be an interesting contribution to the community.

---

### Official Review · Reviewer_K8Hd · 2021-07-16

**Rating:** 7
**Confidence:** 2

**Summary:**

This paper tackles the problem of learning representations for imitation learning, in the context of offline data (with demonstrations). Firstly, given some low dimensional paramterization of the state, the authors provide a performance bound on the learnt policy over the representation matching the ground truth policy, regardless of the structure of the ground truth, as well as bound on the sample efficiency. The authors show that it is possible to learn the model and representations using a contrastive learning objective, and the final representation is output by applying random fourier features to the features learnt via contrastive learning. The authors provide some toy experiments in tabular decision tree settings to verify this hypothesis as well as on offline Atari dataset.

**Limitations And Societal Impact:**

the authors do not address limitations of their work in detail and do not have a section on potential negative societal impact.

**Main Review:**

Strengths: This paper tackles representation learning in offline data and 1) presents a thorough proof of the optimal representations needed for imitation learning, 2) the contrastive objective based representation learning insights seem very useful as well. The bounds derived seem quite thorough and the experiments provided are informative. The paper is overall clear and well written.


Weaknesses: While the paper provides potentially useful insights, the fact that most cases consider linear dynamics makes it hard to be applicable to realistic settings. It would be good for the authors to discuss which aspects/assumptions make it difficult to scale to the real world, and potential down-stream uses of the learnt representations/policies, especially since the authors provide experiments on a non-standard Atari setting. It would also be good if the authors provided an exact training procedure used to learn the representation from the contrastive objective.

Clarifications:

What term in theorem 3 shows the dependency on the quality of the representation?

Why is theorem 4 only in the tabular case? How does learning happen in tabular cases?



**Time Spent Reviewing:**

5

---

> ### Author Response · Authors · 2021-08-09
> **Response**
>
> We thank the reviewer for their positive assessment. We address the reviewer’s feedback below:
>
> “the fact that most cases consider linear dynamics makes it hard to be applicable to realistic settings”
>
> We respectfully disagree with this characterization. First, we want to emphasize that only Theorem 3 considers linear dynamics (Theorem 2 applies to any approximate dynamics models), and even in Theorem 3 the only reliance is that the MDP can be *approximately* represented by linear dynamics. Second, we disagree that this reliance on linear models limits the applicability of our method. Through the idea of “contrastive Fourier features” in our paper, we have already presented one way in which highly expressive dynamics models can be parameterized in a linear way. In a similar vein, much work in the NTK literature shows that large neural networks in general can be expressed as linear models (see “​​Wide Neural Networks of Any Depth Evolve as Linear Models Under Gradient Descent” by Lee, et al.). Thus, we believe that linear dynamics should not severely limit our method’s applicability to realistic settings.
>
> “It would be good for the authors to discuss which aspects/assumptions make it difficult to scale to the real world…”
>
> Thanks for the suggestion. We will be sure to include an elaboration on the limitations of the work -- e.g., issues with too much of an offline distribution shift, or potential hardness of actually learning a near-optimal representation $\phi$ -- in the final version of the paper.
>
> “It would also be good if the authors provided an exact training procedure…”
>
> A pseudocode is provided in Appendix A of the supplementary materials. We will expand on this pseudocode to provide a more detailed description of the algorithm.
>
> “What term in theorem 3 shows the dependency on the quality of the representation?”
>
> In Theorem 3, the quality of the representation is given by the 1st term on the right-hand-side, specifically the quantity \epsilon_{R,T}. This is the same term that appears in Theorem 2.
>
> “Why is theorem 4 only in the tabular case? How does learning happen in tabular cases?”
>
> Theorem 4 focuses on formalizing the claim that “a smaller representation should provably improve sample-efficiency of downstream BC.” We consider only the tabular case in Theorem 4 to simplify exposition (a non-tabular setting would require more complicated dependence on the parameterization of the policy and the properties of the learning algorithm). For learning in the tabular case, we assume the learned latent policy $\pi_{N,Z}$ is learned to exactly minimize $J_{BC,\phi}$; in other words, $\pi_{N,Z}(a|z)$ is the empirical conditional distribution appearing in $D^{\pi^*}_N$ with respect to the representation $\phi_M$.

---

### Official Review · Reviewer_wHHg · 2021-07-17

**Rating:** 7
**Confidence:** 4

**Summary:**

The authors study the impact of representation learning when applied to behavioral cloning (BC), and present a series of interesting theoretical results that establish its utility in this context. In particular, the authors show that there is a direct relationship between the quality of the state representation and the performance of the policy derived using BC over that representation. The authors then present a practical representation learning algorithm designed using the insight gained from their theoretical results and show that using it in combination with BC results in an improvement over comparison approaches.

**Limitations And Societal Impact:**

I found the authors’ treatment of these topics in the paper to be sufficient.

**Main Review:**

STRENGTHS:

(S1) The overall thrust of the work, i.e., the studying the connection between representation learning and imitation learning, is excellent and extremely important to the community.

(S2) The paper contains interesting and seemingly well-founded theoretical insights that have practical implications.

(S3) The authors have proposed a practical algorithm based on their analysis that appears to achieve good empirical results.

WEAKNESSES:

(W1) My major concern with the paper is that it seems to ignore a major portion of the literature that has provided the field with an understanding of behavioral cloning (BC) for the better part of a decade, i.e., the work of Stephane Ross and Drew Bagnell as far back as 2010 (e..g, [Ross and Bagnell, AISTATS 2010]). In the submission under review here, the authors seem to identify the major problem with BC to be that of aliasing, whereas in the referenced work above—and throughout the community, to the best of my own knowledge—the major issue is commonly understood to be the quadratic growth in error with respect to the task duration. While both issues seem to be intrinsically linked to a lack of demonstration data, it’s not clear the extent to which the ignorance of this work impacts the presented technique. Is it simply a hole in the related work which the authors can easily address, or is it a more fundamental issue? For example, in the error bounds derived by the authors, I do not see any hint of an error term that grows quadratically with the duration of the task—where have these terms gone? I am very curious to read the authors’ response.


POST-DISCUSSION COMMENTS:

Thanks to the authors for addressing (W1) above. As mentioned in the discussion, I’d like to see some text added to the next version of the paper acknowledging this connection and explicitly drawing the distinction between distribution shift and aliasing.

**Time Spent Reviewing:**

3

---

> ### Author Response · Authors · 2021-08-09
> **Response**
>
> Thank you for the close reading of our paper! We are happy that you generally found the work’s contributions excellent and impactful. We address the concerns regarding related work below. Please let us know if these comments adequately address all your concerns with the paper or if you have any further questions.
>
> “authors seem to identify the major problem with BC to be that of aliasing”
>
> We respectfully disagree with this characterization. We do not claim that aliasing is a major problem with BC, but rather aliasing is a major problem specifically when *representation learning* is performed in conjunction with downstream BC. Traditionally -- i.e., in the works of Ross & Bagnell and others -- the aliasing issue is ignored because it is irrelevant in their considered settings, which perform BC from the raw MDP observations. Indeed, when performing BC from the raw MDP observations there is, by definition of the MDP, no aliasing. The aliasing issue only arises when one performs BC w.r.t. some devised representations of the raw observations, which is a desirable approach when the raw observation space is very large compared to the number of demonstrations available. This is the setting we focus on in our work.
>
> “in the error bounds derived by the authors, I do not see any hint of an error term that grows quadratically with the duration of the task—where have these terms gone?”
>
> The quadratic dependence on task horizon appears in the form of $(1-\gamma)^{-2}$ in all bounds. In Lemma 1 (which is a discounted variant of the same result in Ross & Bagnell 2010), this is in the coefficient $Rmax/(1-\gamma)^2$ of Eq 7. In Theorem 2, this is in the constant C of Eq 11. In Theorem 3, this is in the constant C of Eq 14.
>
> Generally, we view aliasing as an orthogonal problem to quadratic dependence on horizon. Indeed, if some representation maps two very different raw observations to the same latent state, no downstream imitation learning algorithm (regardless of sample complexity) will be able to learn a good policy.
>
> We focus our present work on BC as the downstream imitation learning algorithm, which has a quadratic dependence on horizon. Despite the simplicity of BC, even for this setting deriving the representation bounds in our paper is highly nontrivial. Extending our representation learning bounds to more sophisticated algorithms with potentially smaller dependence on horizon, like DAgger (Ross/Gordong/Bagnell 2011), is a promising direction for future work.

---

> > ### Comment · Reviewer_wHHg · 2021-08-31
> > **Thanks**
> >
> > Thanks to the authors for addressing my question and pointing out where the quadratic dependence appears. I’d really like to see this point addressed in the next revision of the paper (ie, explicit acknowledgement of the distribution shift problem from Ross/Bagnell and a hint to the reader where this appears)—the way the authors explained it above is very good and should be the basis for that revision.
> >
> > Trusting the authors to do that, I’m comfortable raising my score.

---

### Official Review · Reviewer_QKhN · 2021-07-20

**Rating:** 7
**Confidence:** 4

**Summary:**

This paper considers using offline datasets for improving representation learning for subsequent imitation learning.

**Limitations And Societal Impact:**

I think the authors need to better address the limitation of the work so that its more informative for practical scenarios, this could for instance include some paragraphs about the specific shortcomings of the method complemented with the theoretical results.

**Main Review:**

The paper is in general well written and the theoretical contributions are nicely explained with important differences from the similar field of bisimulation metric based methods. The authors also validate the approach on toy environment and Atari domain, although I would suggest the authors to also report percentile performance improvements for individual Atari games which seem to be missing in the report. Overall the paper is a decent contribution to the field of imitation learning.

**Time Spent Reviewing:**

8

---

> ### Author Response · Authors · 2021-08-09
> **Response**
>
> Thanks for taking the time to read our submission and we are happy that you enjoyed it. We appreciate your feedback and suggestions. Regarding performance per individual Atari game, yes we will be sure to include this in the final version of the paper. Regarding more detailed discussions of limitations, yes, we will also be sure to include an elaboration on the limitations of the work -- e.g., issues with too much of an offline distribution shift, or potential hardness of actually learning a near-optimal representation $\phi$ -- in the final version of the paper.

---

### Decision · Program_Chairs · 2021-09-27

**Decision:**

Accept (Poster)

**Comment:**

After discussion, all reviewers believe that the rebuttal has addressed their concerns and agree on an acceptance.